# Transcriptome Analysis of Macrophytes’ *Myriophyllum spicatum* Response to Ammonium Nitrogen Stress Using the Whole Plant Individual

**DOI:** 10.3390/plants12223875

**Published:** 2023-11-16

**Authors:** Wyckliffe Ayoma Ochieng, Li Wei, Godfrey Kinyori Wagutu, Ling Xian, Samuel Wamburu Muthui, Stephen Ogada, Duncan Ochieng Otieno, Elive Limunga Linda, Fan Liu

**Affiliations:** 1Core Botanical Gardens/Wuhan Botanical Garden, Chinese Academy of Sciences, Wuhan 430074, China; ochiengwycliffe5@gmail.com (W.A.O.); gkwagutu@gmail.com (G.K.W.); xianling@wbgcas.cn (L.X.); smuthui41@gmail.com (S.W.M.); otienod45@yahoo.com (D.O.O.); 2Sino-Africa Joint Research Centre, Chinese Academy of Sciences, Wuhan 430074, China; 3University of the Chinese Academy of Sciences, Beijing 101408, China; 4Changjiang Water Resources and Hydropower Development Group (Hubei) Co., Ltd., Wuhan 430010, China; weili19810816@outlook.com; 5Institute for Biotechnology Research, Jomo Kenyatta University of Agriculture and Technology, Nairobi 00200, Kenya; stevogada@gmail.com; 6School of Resources and Environmental Science, Hubei University, Wuhan 430062, China; limungaelive@gmail.com

**Keywords:** ammonium stress, submerged macrophytes, *Myriophyllum spicatum*, gene expression, transcriptome

## Abstract

Ammonium toxicity in macrophytes reduces growth and development due to a disrupted metabolism and high carbon requirements for internal ammonium detoxification. To provide more molecular support for ammonium detoxification in the above-ground and below-ground parts of *Myriophyllum spicatum*, we separated (using hermetic bags) the aqueous medium surrounding the below-ground from that surrounding the above-ground and explored the genes in these two regions. The results showed an upregulation of asparagine synthetase genes under high ammonium concentrations. Furthermore, the transcriptional down and/or upregulation of other genes involved in nitrogen metabolism, including glutamate dehydrogenase, ammonium transporter, and aspartate aminotransferase in above-ground and below-ground parts were crucial for ammonium homeostasis under high ammonium concentrations. The results suggest that, apart from the primary pathway and alternative pathway, the asparagine metabolic pathway plays a crucial role in ammonium detoxification in macrophytes. Therefore, the complex genetic regulatory network in *M. spicatum* contributes to its ammonium tolerance, and the above-ground part is the most important in ammonium detoxification. Nevertheless, there is a need to incorporate an open-field experimental setup for a conclusive picture of nitrogen dynamics, toxicity, and the molecular response of *M. spicatum* in the natural environment.

## 1. Introduction

Aquatic plants are fundamental components affecting food webs and functions in aquatic ecosystems [1,2]. Submerged macrophytes improve water quality by sequestering nutrients and suppressing sediment suspensions [3,4]. Plants (both terrestrial and aquatic) are considered under stress when environmental conditions are not ideal for growth and survival. How adverse environments affect plant growth, survival, and metabolism and how plants respond to stressors is a fundamental scientific question that is vital for biodiversity, restoration, and agriculture [5,6]. Nitrogen is quantitatively the most crucial nutrient for plants and a major restricting factor in plant productivity. Plants take up mainly three forms of nitrogen as follows: nitrate, ammonium, and organic nitrogen. As the primary nitrogen source, ammonium is significant for plant growth and development in freshwater biological systems. Moreover, the lower energy requirement for its assimilation makes it the preferred nitrogen source compared to nitrate (NO_3_^−^) [7]. However, a high ammonium concentration harms aquatic plants [8,9].

Macrophytes manage ammonium stress through varied detoxification mechanisms. Ammonium is assimilated into the glutamine and glutamate in plant cells, which is catalyzed by the primary pathway through glutamine synthetase (GS) and glutamate synthase (GOGAT) and the alternative pathway through glutamate dehydrogenase (GDH) [10,11]. Furthermore, apart from the primary pathway and alternative pathway, the asparagine metabolic pathway through cytosolic asparagine synthetase (AS) is also essential in ammonium assimilation [12,13]. Asparagine synthetase plays an important role in plant nitrogen assimilation and distribution. Asparagine, the product of asparagine synthetase, is one of the main reserve compounds responsible for organic nitrogen transport and storage in plants due to its high nitrogen/carbon ratio. It is, thus, a key enzyme in nitrogen assimilation in higher plants [14]. In previous studies, the expression level of the gene encoding the enzyme is related to nitrogen in different forms. For instance, ammonium induces the Arabidopsis thaliana AS gene, increasing its expression level [15]. It is also induced by nitrate in *Phaseolus vulgaris* and soybean. Furthermore, in Poplar trees and wheat, asparagine is the major nitrogen transport compound that plays a role in transporting nitrogen between the sinks and source tissues [14,16,17]. Moreover, a physiological study recorded an increase in asparagine synthetase enzyme activity with an increase in ammonium concentration in *Myriophyllum aquaticum* [11]. However, for *Myriophyllum spicatum*, no other pathway has been recorded for nitrogen metabolism besides the primary ammonium assimilation pathway and the alternative pathway. Therefore, due to the complex nature of the carbon/nitrogen network and metabolism, there is a need to further explore this macrophyte species and understand better the genes, pathways, and detoxification mechanisms in different tissues.

Next-generation sequencing technology (RNA-seq) has been used in recent years on *Myriophyllum aquaticum*, *Zostera marina*, rice (*Oryza sativa*), and the aquatic plant duckweed (*Lemna minor*) to reveal this molecular mechanisms’ response to high NH_4_^+^ concentrations at the transcriptional level [18,19,20,21,22,23]. This, therefore, provides the idea behind exploring high NH_4_^+^ tolerance mechanisms in *M. spicatum*. The most recent study by [24] on *Myriophyllum aquaticum* explained the essential role of the transcriptional up-regulation of light-harvesting chlorophyll *a*/*b*-binding protein genes in leaves toward high NH_4_^+^ resistance.

This study aimed to explore the crucial genes involved in ammonium stress in the above-ground and below-ground regions of *M. spicatum* and examine the related pathways for ammonium detoxification. This study gives baseline data on molecular toxicological mechanisms in response to ammonium, providing important insights into the further genomic analyses of the biological and molecular mechanisms of ammonium stress and accumulation in aquatic plants.

## 2. Materials and Methods

### 2.1. Plant Material Growth

*Myriophyllum spicatum* was chosen as the experimental material because of its tolerance to high ammonium concentrations, as illustrated in previous research [25,26]. Plant materials were randomly collected from the same colony in Xingyun Lake, a freshwater lake (24.33° N, 102.79° E, ca 1740 m above sea level) adjacent to sub-basins in the central Yunnan plateau in Southwest China. Xingyun Lake is eutrophic, with heavy cyanobacterial blooms throughout the year [26].

### 2.2. Experimental Design

Approximately 10 cm apices of healthy materials were grown in small pots (0.15 m deep and 0.26 m diameter) and put into big tanks (0.75 m deep and 0.45 m diameter) in the open ground. After four months of cultivation, healthy plants (whole plants) were collected for laboratory experiments. The materials (15 g for each bowl) were pre-cultured (acclimatized) in distilled water for 24 h before exposure to ammonium concentrations. Four ammonium concentrations (0.1 mg/L, 3 mg/L,15 mg/L, and 50 mg/L) were used for the treatment of the shoot (leaves and stem). This is because 0.1 mg/L is the normal ammonium concentration in lakes; 3 mg/L is related to high ammonium in lakes; 15 mg/L is related to water bodies; and 50 mg/L is the highest concentration that most macrophytes can thrive in [25,27]. For the roots, 2.5 mg/L, 75 mg/L, 375 mg/L, and 1250 mg/L ammonium concentrations were used, which were set according to the ratio 25:1 below-ground/above-ground [28]. The concentrations and symbols are summarized in Table 1. Sealed water-tight hermetic bags separated ammonium nutrient solutions for the above and below-ground regions. The below-ground regions of the macrophyte were gently placed in the hermetic bag containing 0.1 L of the below-ground nutrient solution and were then tightly tied to prevent the two solutions from mixing. The hermetic bag and its contents (roots and ammonium solutions) were covered with aluminum foil to exclude light and were immersed in bowls (culture pots) containing 1.25 L of the nutrient solution for the above-ground region. The ammonium solutions were prepared through the addition of (NH_4_^+^)_2_SO_4_ to Hoagland’s solution, pH was adjusted to 8.0 ± 0.10 [25,26], and three replicates were used for each concentration. The controlled environmental conditions were set at a temperature of 25 °C, with a photoperiod of 14/10 h (light/dark) and light intensity of 108 μmol photons m^−2^ s^−2^ [25,26]. After 4 days of culturing in ammonium concentrations, samples were collected for RNA extraction. Approximately 0.5 g of the plant material (roots, leaves, and stem) for each concentration was collected and frozen immediately in liquid nitrogen, placed in dry ice, and sent to the company (Frasergen, Shanghai, China) for RNA extraction.

### 2.3. RNA Extraction, cDNA Library Construction, and Transcriptome Sequencing

The total RNA from each sample was extracted using a TRIzol reagent (Invitrogen, Carlsbad, CA, USA) according to the manufacturer’s instructions. The quality and quantity of the total RNA were assessed using 1% agarose gel and examined with a NanoDrop 2000 c spectrophotometer (NanoDrop Technologies, Wilmington, DE, USA). The RNA integrity number (RIN) was assessed using an Agilent 2100 Bioanalyzer (Santa Clara, CA, USA). The RNA was used for subsequent Illumina library preparation if the RIN was greater than 8.0. The total RNA was stored at −80 °C.

The full-length cDNA of mRNA was synthesized using the Clonetech SMARTer PCR cDNA Synthesis Kit (Illumina Inc., San Diego, CA, USA) and subjected to PacBio single-molecule real-time sequencing according to the manufacturer’s protocol. Primers with Oligo dT were used to pair A-T bases with poly-A as primers for the reverse synthesis of cDNA, and primers were added to the end of the full-length cDNA of reverse synthesis. Full-length cDNA was amplified via a PCR, and the product was purified using PB magnetic beads to remove some small fragments of cDNA less than 1 kb. The unconnected fragments were digested via exonuclease and purified using PB magnetic beads to obtain the sequencing library. After the library construction was completed, Qubit 3.0 was used for accurate quantification, and Agilent 2100 was used to detect the library size (Agilent Technologies, Palo Alto, CA, USA). Sequencing was only carried out after the library size met expectations (Frasergen, Shanghai, China).

### 2.4. De Novo Transcriptome Assembly and Annotation

The raw reads were cleaned by removing the low-quality adapter and read with 5% or more unknown nucleotides. The obtained quality-filtered reads were, de novo, assembled into contigs via the Trinity Program [29]. The putative functions of unigene sequences were annotated using BLASTx (E-value ≤ 10^−5^) with several protein databases (euKaryotic Orthologous Groups (KOG), Gene Ontology (GO) and Kyoto Encyclopedia of Genes and Genomes (KEGG). ESTScan was used to determine its sequencing direction when a unigene could not be aligned to any of the above databases. Blast2GO 2.5.0 [30] was employed to compare and determine the unigene Gene Ontology (GO) annotations.

### 2.5. Reads Mapping and Analysis of Differentially Expressed Genes

RNA-Seq Analysis Snake Make Workflow (RASflow) [31] was used for these analyses. In brief, the reads for each sample were aligned onto the annotated *M. spicatum* transcriptome using HISAT2 [32]. Furthermore, we mapped the reads to transcriptome through the use of the pseudo alignment with Salmon [33] to obtain a quantified table of the transcripts’ expression. This was proceeded with feature counting using feature counts [34]. The raw counts were normalized into estimated Transcripts Per Million (TPM) from Salmon, which were scaled using the average transcript length over samples and the library size via tximport [35]. Expression abundance values (read counts) for each group (leaf, stem, and root) and their various concentrations (Control, T1, T2, T3, and T4) were normalized using DEseq2 implemented in iDEP for differential gene analysis [36]. DESeq2 was used for differential expression significance analysis between the control (CK0) and treatments (for the leaf, root, and stem) under the screening threshold of FDR (false discovery rate) < 0.05, log2FC (fold change) >1 or <−1 [37].

### 2.6. Identification and Construction of Co-Expressed Gene Networks

We used the package WGCNA [38] implemented in iDEP [36] to construct a co-expressed gene network using the normalized abundance values. The most highly variable genes (top 1000) with a soft threshold of 12 and a minimum module size of 20 genes were selected.

## 3. Results

### 3.1. Basic Information on the Transcriptome Sequence

An overview of the RNA-Seq reads derived from the sequencing is presented in Table 2, Table 3 and Table 4 for the leaf, stem, and root, respectively. In total, approximately 633 million, 695 million, and 578 million clean reads were obtained from the leaf, stem, and root, with an average GC content of 46.04%, 45.60%, and 48.34%, respectively (Table 2, Table 3 and Table 4). In addition, the higher Q20 (94.84%, 95.02%, and 94.95% in leaf, stem, and root, respectively) and Q30 values (86.67%, 87.27%, and 87.03% for the components) indicated the high quality of the transcriptome sequencing (Table 2, Table 3 and Table 4).

### 3.2. Overlaps of Differentially Expressed Genes in Leaf, Stem, and Roots after Exposure to Ammonium Treatments

We recorded different expressions of genes in the plant parts, that is, the leaf, stem, and root. Notably, the differentially expressed genes were higher in high ammonium concentrations in the leaf and stem. Furthermore, the most differentially expressed genes (DEGs) were between the control and highest ammonium concentrations for all plant parts. However, the root had an exceptionally high number of overlapping DEGs for all concentrations compared to the stem and leaf (Figure 1a–c).

### 3.3. K-Mean Clusters of Differentially Expressed Genes and GO Enrichment of Most Variable Genes

Two thousand (2000) of the most variable genes were clustered into three major GO biological process categories; these included response to hypoxia, biosynthetic process, and photosynthesis. Cluster A had 617 genes (higher expression in roots compared to leaf and stem), Cluster B had 831 genes, and Cluster C had 552 genes (higher expression in stem and leaf compared to roots). The enrichment tree showed the pathways involved (Figure 2a,b).

### 3.4. Principal Component Analysis (PCA) of the Genes

Principle component analysis was used to explain the expression of the genes in the whole plant. It showed that the components above-ground (leaf and stem) and below-ground (root) were distinct, as illustrated on the first axis of PCA, which explained the 43% variation among the plant components and concentrations. The second PCA axis explained a 10% variation among the plant parts and concentration (Figure 3).

### 3.5. Expression Modules

From the top 1000 genes used for module extraction on iDEP, a total of four modules were generated (Figure 4a,b). These included module 1 (467 genes), module 2 (126 genes), module 3 (117 genes) and module 4 (48 genes). We selected module 3 because it included more target genes related to ammonium stress (Figure 4a).

### 3.6. Expression of the Top 10 Genes from the Selected Module

A comparison of gene expressions for the top 10 genes in the three components (leaf, stem, and root) using counts per million was conducted (CPM). This was obtained by normalizing the read counts by the total counts per sample. The top 10 genes included AATC (aspartate aminotransferase), AMT12 (ammonium transporter 1 member 2), ASPGB1 (L-asparaginase), STY46 (serine/threonine-protein kinase), GLT1 (Glutamate synthase 1 [NADH]), ACA1 (alpha carbonic anhydrase), GLU1(Ferredoxin-dependent glutamate synthase 1), DHE4(NADP-specific glutamate dehydrogenase), ASNS (asparagine synthetase [glutamine-hydrolyzing]), and GLNA1(glutamine synthetase cytosolic isozyme 1). Notably, the expressions of asparagine synthetase and serine/threonine-protein kinase genes increased with the increase in ammonium concentration in the leaf and stem (Figure 5A,B). Generally, the trends in the expression of genes in the leaf and stem were similar. There was no significant trend in the expression of the genes in the root (Figure 5C).

### 3.7. A Comparison of Gene Regulations between the Above-Ground and Below-Ground Parts

Notably, asparagine synthetase (ASNS) and serine/threonine-protein kinase STY46 (STY46) (Appendix A) genes were upregulated in both the above-ground and below-ground, with more upregulation occurring in the above-ground (Figure 6). Furthermore, in the above-ground part, specifically leaf 3, more genes were upregulated, including glutamate dehydrogenase, ferredoxin-dependent glutamate synthase 1, and aspartate aminotransferase were found at a high ammonium concentration (50 mg/L). However, in the below-ground part, two more genes, L-asparaginase (asparaginase) (Figure 6) and alpha carbonic anhydrase (α CA) (Appendix A), were upregulated at a high ammonium concentration, while others were downregulated.

## 4. Discussion

### 4.1. Response of Genes Related to Ammonium Assimilation

In higher plants, ammonium originates from nitrate reduction, direct absorption, photorespiration, gaseous nitrogen (N_2_) fixation, or the deamination of nitrogenous compounds, such as asparagine [39]. All inorganic nitrogen is first reduced to ammonium because it is the only reduced nitrogen form available to plants for assimilation into N-carrying amino acids [40]. The incorporation of ammonium into the pool of N-containing molecules is first catalyzed by the glutamine synthetase (GS)/glutamate synthase (GOGAT) cycle [41]. However, previous research found that an alternative pathway (GDH) is also important and used by plants to detoxify ammonium under high concentrations [25,42,43,44,45]. Apart from the GS/GOGAT cycle and GDH, cytosolic asparagine synthetase (AS) is also important in ammonium assimilation [12,13]. The downregulation of NADH-GOGAT genes in all the treatments for the above-ground and below-ground in high ammonium concentrations and the insignificant regulation of Fd-GOGAT suggests that GOGAT played no role in ammonium detoxification. This is also displayed in the regulation of GS genes. Therefore, the overall inhibition of the GS-GOGAT cycle suggests that this pathway was not the main approach for NH_4_^+^ detoxification in *M. spicatum.* However, the upregulation of GDH in high concentrations (50 mg/L), specifically for the leaf, suggests that the alternative pathway is crucial for ammonium assimilation. This is consistent with a previous study by [25], which found the important role played by GDH enzymes in ammonium detoxification for this species.

The upregulation of asparagine synthetase genes with an increase in the ammonium concentration of the leaf and stem (15 mg/L and 50 mg/L) and root (375 mg/L and 1250 mg/L) suggests that asparagine synthetase played a role in ammonium detoxification by aiding the recycling of excess ammonium in the tissues into asparagine. Previous studies show that asparagine accumulates under high levels of exogenous ammonium [15] and that asparagine synthetase could play a role in ammonium detoxification and distribution [46,47]. Furthermore, an increase in the expression levels of AS with a rise in ammonium concentration has been recorded in *Arabidopsis thaliana* [15]. Moreover, Ref. [11] recorded the close association of AS in the detoxification of ammonium toxicity in *Myriophyllum aquaticum*. Aspartate biosynthesis is mediated by aspartate aminotransferase (AspAT), which catalyzes the reversible transamination between glutamate and oxaloacetate to generate aspartate and 2-oxoglutarate via a double displacement kinetic mechanism [48] and is also essential for the production of malate, which is needed in the mitochondria for the tricarboxylic acid (TCA) cycle [49]. Furthermore, AspAT plays a role in the primary ammonium pathway by recycling carbon skeletons for transamination [50]. Therefore, the upregulation of AspAT in the leaf, specifically at a high ammonium concentration, suggests part of the mechanism for competitiveness and tolerance of this macrophyte species to ammonium by providing the energy needed for ammonium assimilation and detoxification.

### 4.2. Response of Genes Related to Carbon Metabolism

Protein kinases constitute one of the largest gene families in plant genomes. These enzymes catalyze the reversible phosphorylation of specific amino acids (serine, threonine, and tyrosine) to regulate the activity of their target proteins [51]. The upregulation of serine/threonine-protein kinase STY46 in high ammonium concentrations of the above-ground and below-ground parts of this present study suggests that this gene plays a role in response to ammonium exposure or stress. This is consistent with a previous study conducted on *Arabidopsis thaliana* by [52], which recorded the involvement of cytosolic protein kinase STY46 on plant growth and abiotic stress responses.

Carbonic anhydrase (CA) exhibits various distribution patterns among organs, tissues, and cellular organelles commensurate with its diverse physiological roles. It has been found in high amounts in the leaves of plants [53,54] and leguminous root nodules [55]. A previous study on *Arabidopsis thaliana* discovered α-CA1 in the chloroplast stroma [56]. The downregulation of α CA in the leaf and stem suggests that CO_2_ is not limited; this is evident in photosynthetic parameters like the F*v*/F*m* and Chlorophyll content, which did not record significant change with an increase in ammonium concentration (not included in the paper). However, the upregulation of α CA in the root suggests an elevated CO_2,_ which could increase root growth to balance nutrient uptake with the rate of sugar production from increased photosynthesis, or the root system, which acts as a sink to store excess sugars [57].

### 4.3. Ammonium Transporter Genes and Glutathione S-Transferase Genes

Generally, ammonium transporter (AMT) family proteins are used by plants to absorb NH_4_^+^ due to their low concentration in the environment [58]. Plant ammonium transport is mediated by the transporter gene families AMT1, AMT2, AMT3, AMT4, and AMT5 [59,60,61]. The current study recorded the downregulation of AMT 1.2 genes in all ammonium concentrations in the above-ground and below-ground. It shows that this ammonium transporter subfamily (AMT1.2) plays no role in the ammonium uptake of this macrophyte species. However, the downregulation can suggest a mechanism by the macrophyte species to avoid the further uptake of ammonium into the above-ground plant tissues, keeping a balanced in vivo ammonium level. This is consistent with previous research performed on *Lotus japonicus* under drought stress [62].

## 5. Conclusions

This study compared the ammonium stress genes of the leaf, stem, and root using whole plant individuals of the rooted and submerged macrophyte *M. spicatum*. Our findings show the important role played by the asparagine metabolic pathway through the asparagine synthetase, asparaginase, and aspartate aminotransferase and the above-ground part in ammonium absorption, assimilation, and translocation. This information, therefore, provides further insights into the ammonium response mechanisms of submerged macrophytes and gives a molecular backup for physiological detoxification mechanisms. Nonetheless, there is a need to incorporate an open-field experimental setup for a conclusive picture of nitrogen dynamics, toxicity, and the molecular response of *M. spicatum* in the natural environment.

## Figures and Tables

**Figure 1 plants-12-03875-f001:**
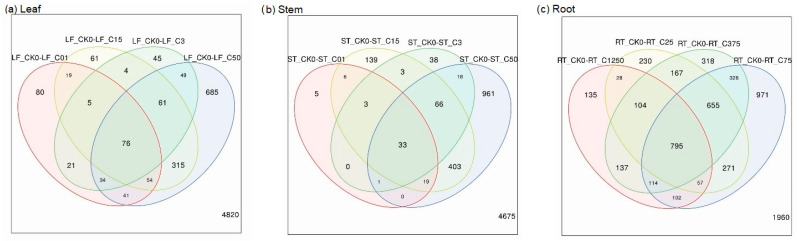
Overlaps of differentially expressed genes in treatments of the leaf (**a**), stem (**b**), and roots (**c**) compared to their respective controls (LF_CK0, ST_CK0, and RT_CK0). The CK0(control), C01, C3, C15, C50 are the ammonium concentrations 0.1 mg/L, 3 mg/L, 15 mg/L and 50 mg/L, respectively, for the leaf and stem, while CK0, C25, C75, C375 and C1250 are the ammonium concentration 2.5 mg/L, 75 mg/L, 375 mg/L and 1250 mg/L, respectively, for the root.

**Figure 2 plants-12-03875-f002:**
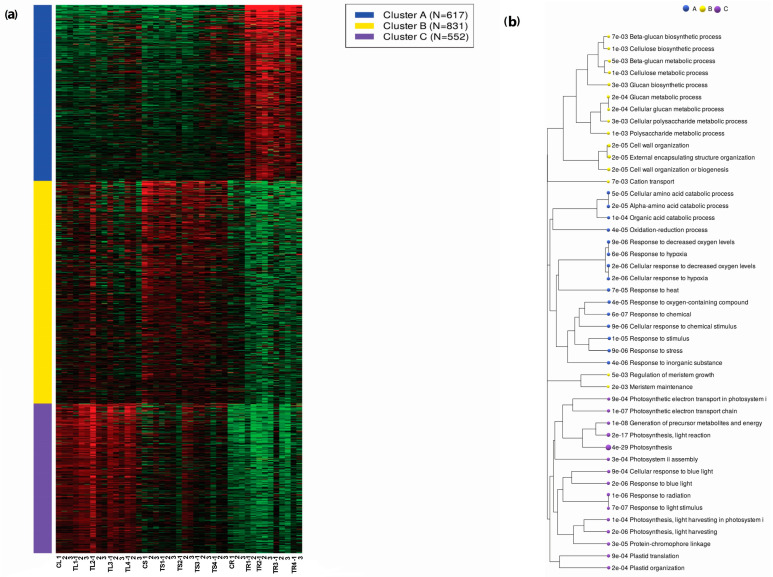
K-means clustering (**a**) and GO enrichment (**b**) for the top 2000 most variable genes. The symbols CL, CS, and CR represent the controls, while TL, TS, and TR represent the treatments in the leaf, stem, and root, respectively. The numbers 1, 2, and 3 represent the replicates. Red color shows high expression while green color low expression. The “e” notations represent the scientific notation, for example, 7e-03 represents 7 × 10^−3^.

**Figure 3 plants-12-03875-f003:**
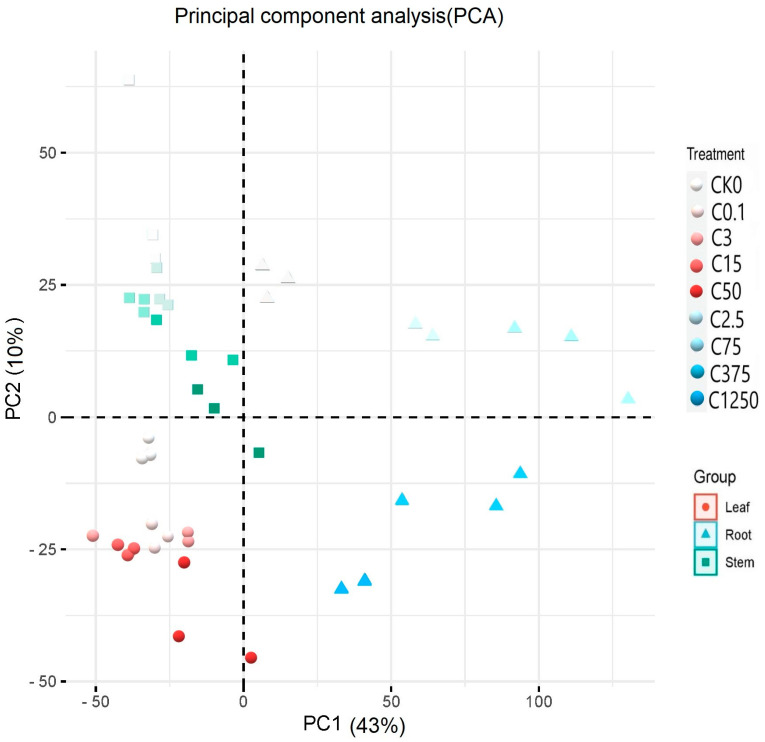
Principal component analysis (PCA) of genes. The red, green, and blue colors represent leaf, root, and stem components, respectively. Different shapes are used to differentiate the plant’s parts. The circle, square, and triangle represent the leaf, stem, and root, respectively. Ammonium concentrations in the respective plant parts are differentiated from lighter (low ammonium concentration) to darker colors (high ammonium concentration).

**Figure 4 plants-12-03875-f004:**
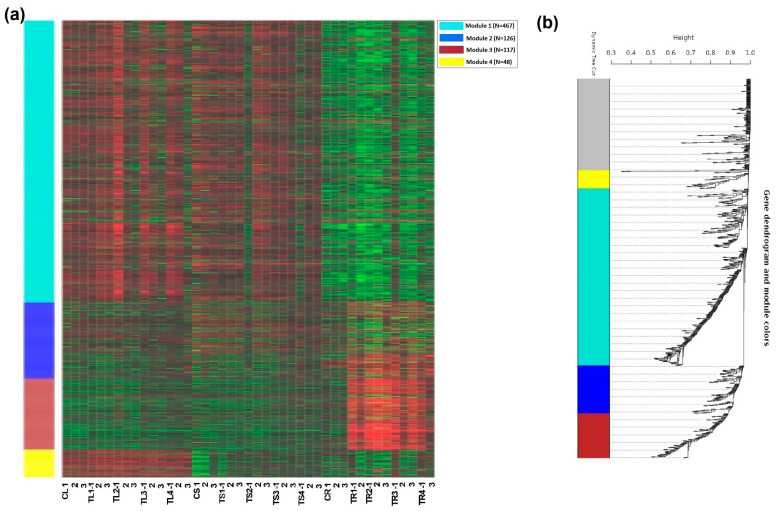
Four modules of co-expressed genes (**a**) based on the 1000 most variable genes were identified using WGCNA (weighted correlation network analysis) (**b**). The symbols CL, CS, and CR represent the controls, while TL, TS, and TR represent the treatments in the leaf, stem, and root, respectively. Red and green colors in (**a**) shows high expression and low expressions respectively. The grey module corresponds to the set of genes which have not been clustered in any module.

**Figure 5 plants-12-03875-f005:**
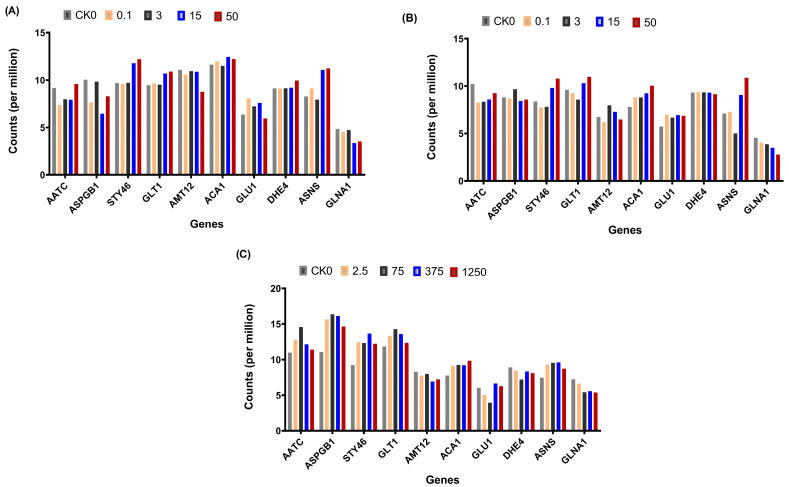
Comparison of gene expression in the top 10 genes in the three components (leaf, stem, and root) using normalized data. The genes included serine/threonine-protein kinase STY46 (STY46), aspartate aminotransferase (AATC), L-asparaginase (ASPGB), alpha carbonic anhydrase (α CA), ammonium transporter 1 member 2 (AMT12), asparagine synthetase (ASNS), glutamate synthase 1 [NADH]) (GLT1), ferredoxin-dependent glutamate synthase 1 (GLU1), NADP-specific glutamate dehydrogenase (DHE4), and glutamine synthetase cytosolic isozyme 1 (GLNA1). (**A**–**C**) represent the leaf, stem, and root, respectively.

**Figure 6 plants-12-03875-f006:**
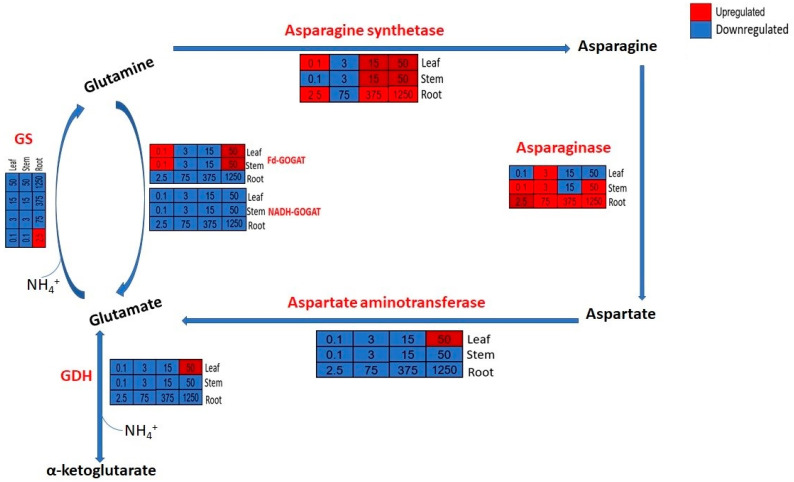
Schematic summary of ammonium assimilation network/pathway. The red, blue, and white colors represent upregulated, downregulated, and no change, respectively.

**Table 1 plants-12-03875-t001:** Ammonium concentration for the above-ground and below-ground regions.

Plant Part	Ammonium Concentrations (mg/L)				
Above-ground	CK0	0.1	3	15	50
Below-ground	CK0	2.5	75	375	1250
Symbol	Control (C)	T1	T2	T3	T4

CK0 represents the control (C), and the numbers represent treatments (T1, T2, T3, T4).

**Table 2 plants-12-03875-t002:** Transcriptome sequencing quality statistics analysis at different ammonium concentrations in the leaf.

Sample	Clean Reads	Base Num_Clean	GC%	>Q30	>Q20	Clean Rate
Leaf_CK0_1	9483174	1365253325	44.96%	85.4%	94.78%	90.29%
Leaf_CKO_2	44039636	6346954385	45.44%	86.86%	95.14%	90.07%
Leaf_CKO_3	46669192	6737538060	45.13%	87.14%	95.27%	90.93%
Leaf_C0.1_1	48367430	7071729005	45.47%	87.27%	94.84%	94.7%
Leaf_C0.1_2	43056516	6279483747	45.97%	86.18%	94.36%	94.41%
Leaf_C0.1_3	48723216	7105614339	45.82%	86.66%	94.58%	94.58%
Leaf_C3_1	46622400	6752015620	46.42%	88.19%	95.64%	90.82%
Leaf_C3_2	43454682	6223190987	45.49%	85.5%	94.62%	89.01%
Leaf_C3_3	7974958	1147249076	45.01%	85.18%	94.7%	90.25%
Leaf_C15_1	51618630	7533928462	46.27%	86.7%	94.58%	94.64%
Leaf_C15_2	50278296	7349072444	46.94%	87.34%	94.82%	94.63%
Leaf_C15_3	43884022	6414982330	47.11%	87.48%	94.89%	94.88%
Leaf_C50_1	45608796	6660489072	47.48%	86.78%	94.58%	93.82%
Leaf_C50_2	56796048	8257059066	46.01%	86.85%	95.06%	92.48%
Leaf_C50_3	46767812	6811573000	47.13%	86.57%	94.7%	93.62%

CK0 is the control (samples placed in deionized water).

**Table 3 plants-12-03875-t003:** Transcriptome sequencing quality statistics analysis at different ammonium concentrations in Stem.

Sample	Clean Reads	Base Num_Clean	GC%	>Q30	>Q20	Clean Rate
Stem_CK0_1	43380046	6273729213	45.62%	88.19%	95.6%	89.73%
Stem_CKO_2	38909370	5581236204	46.15%	86.4%	94.85%	86.43%
Stem_CKO_3	48699588	7045460726	45.25%	87.84%	95.54%	90.58%
Stem_C0.1_1	44618368	6528565752	45.34%	87.59%	94.98%	94.75%
Stem_C0.1_2	52092526	7626597011	45.36%	87.74%	95.04%	94.92%
Stem_C0.1_3	40865556	5973232795	45.19%	87%	94.74%	94.95%
Stem_C3_1	44015712	6349464797	44.84%	87.09%	95.25%	90.95%
Stem_C3_2	43578078	6229285621	45.45%	85.24%	94.49%	89.44%
Stem_C3_3	43623748	6311383774	45.48%	87.86%	95.54%	91.16%
Stem_C15_1	52479126	7679519421	45.71%	87.48%	94.92%	94.97%
Stem_C15_2	50737878	7416480012	45.89%	87.26%	94.84%	94.84%
Stem_C15_3	50729090	7415387552	45.87%	87.22%	94.8%	94.59%
Stem_C50_1	48614830	7103220369	45.94%	87.8%	95.24%	94.34%
Stem_C50_2	47811050	6985103878	46.19%	87.07%	94.74%	94.54%
Stem_C50_3	45180526	6602860020	45.71%	87.24%	94.83%	95%

CK0 is the control (samples placed in deionized water).

**Table 4 plants-12-03875-t004:** Transcriptome sequencing quality statistics analysis at different ammonium concentrations in the root.

Sample	Clean Reads	Base Num_Clean	GC%	>Q30	>Q20	Clean Rate
Root_CK0_1	9832836	1419104428	45.4%	86.14%	95.04%	90.24%
Root_CKO_2	40122732	5727077094	45.28%	85.02%	94.36%	87.59%
Root_CKO_3	39167564	5595243942	44.87%	85.16%	94.42%	87.9%
Root_C2.5_1	61172098	8957002394	47.34%	87.02%	94.64%	94.24%
Root_C_2.5_2	55210796	8064235149	48.88%	86.3%	94.26%	94.02%
Root_C75_1	46046854	6791496790	49.27%	90.36%	96.76%	94.43%
Root_C75_2	44461536	6558048748	49.56%	90.2%	96.69%	93.76%
Root_C75_3	8530548	1229173572	47.53%	85.92%	94.91%	90.14%
Root_C375_1	47354218	6900194618	50.47%	85.42%	93.8%	93.56%
Root_C375_2	51698210	7554479584	49.22%	86.34%	94.26%	94.28%
Root_C375_3	48025588	7017901998	46.73%	86.42%	94.4%	94.8%
Root_C1250_1	53495218	7844730098	47.57%	87.54%	94.88%	94.86%
Root_C1250_2	20264096	2954814344	52.79%	89.72%	96.4%	63.52%
Root_C1250_3	53383328	7810306178	51.9%	86.84%	94.48%	93.34%

CK0 is the control (samples placed in deionized water).

## Data Availability

Data are contained within the article and Appendix A.

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
