# Peer review of "Transcriptome Analysis of Macrophytes’ Myriophyllum spicatum Response to Ammonium Nitrogen Stress Using the Whole Plant Individual"

_plants, 2023, doi:10.3390/plants12223875_

Round 1

Reviewer 1 Report

Comments and Suggestions for Authors

Thanks for submitting this work. Although this is done in a controlled environment- a weakness of the study.  The study in itself has novelty. Thus my endoresement for publication after a very minor revision.

In your conclusion section and Abstract, it would be better to emphasize the need for verifcation under open field conditions where ammonium N could be rapidly converted into nitrate. This provides bigger picture of N dynamics, N toxicity and the molecular response of Myriophyllum spicatum in natural environment.

Author Response

Comment 1:

“Thanks for submitting this work. Although this is done in a controlled environment- a weakness of the study.  The study in itself has novelty. Thus, my endorsement for publication after a very minor revision”.

Response:

  Thank you for the positive comment. Yes, in the future, we plan to incorporate open-field setups.

Suggestion:

In your conclusion section and Abstract, it would be better to emphasize the need for verification under open-field conditions where ammonium N could be rapidly converted into nitrate. This provides a bigger picture of N dynamics, N toxicity, and the molecular response of Myriophyllum spicatum in the natural environment.

Response:

  Thank you for the important suggestion. We have revised the two sections as suggested.

Reviewer 2 Report

Comments and Suggestions for Authors

Thank you for providing the opportunity to review this manuscript which focused on exploring crucial genes involved in ammonium stress in the above and below ground regions of M. spicatum and to examine the related pathways for amomum detoxification. Overall, I felt the study design was well crafted, the results were presented well, and the manuscript was reasonably well-written but English written style needs clean up. I have some specific comments listed below:

Line 37: “Transforming the nutrition and toxic substances” is not clear as written.

Line 38: Clear water is not always a sign of good water quality.

Line 46: Say “Moreover”, not “Besides.”

Line 49: Say “Macrophytes manage ammonium…”

Line 63: What is “Polar”, the tree?

Lines 69-70: These sentences are not clear.

Line 154: What is “pseudo alignment of Salmon..”

Table 1: What is “T1, T2, T3, T4?

Table 2: Condense Table 2 down to 1 page.

Line 212: Say “…component analysis was used to explain the..”

Figure 4, B: Unreadable

Lines 233-235: Sentences are confusing as written.

Figure 6: The y-axes are count of what? Explain. Also, explain what 2.5, 75, 375 and 1250 are in the table legend.

Line 256 and earlier: Explain what “upregulation” means.

Conclusion: These conclusions do not conclude much. Provide explanation for some of the results.

Comments on the Quality of English Language

The manuscript was reasonably well-written but English written style needs clean up.

Author Response

Line 37: “Transforming the nutrition and toxic substances” is not clear as written.

Response: We have revised the sentence accordingly. Thank you

Line 38: Clearwater is not always a sign of good water quality.

Response: We have revised the sentence according. Thank you

Line 46: Say “Moreover”, not “Besides.”

Response: It has been replaced accordingly. Thank you

Line 49: Say “Macrophytes manage ammonium…”

Response: It has been edited accordingly. Thank you

Line 63: What is “Polar”, the tree?

Response: Yes, Poplar tree. We have added “tree” for clarity. Thank you

Lines 69-70: These sentences are not clear.

Response: We have revised the sentence for clarity. Thank you

Line 154: What is “pseudo alignment of Salmon..”

Response: Thank you for the comment. The sentence was actually to read “pseudo alignment with Salmon” and not “pseudo alignment of Salmon”, we are sorry for the error. We have revised it accordingly for clarity and added the reference.

Pseudo-alignment is a method for estimating gene or transcript abundance in RNA-seq data

Table 1: What is “T1, T2, T3, T4?

Response: T1, T2, T3, T4 represent Treatment 1, Treatment 2, Treatment 3, Treatment 4. All these are summarized in Table 1. We are sorry it was erroneously moved to the results section during formatting. We have revised the sentences (Lines 104- 105) and moved the table back to the methodology section for clarity. Thank you

Table 2: Condense Table 2 down to 1 page.

Response: Thank you for the comment. We have condensed all the tables to fit in pages because previously before formatting, they were in landscape orientation.

Line 212: Say “…component analysis was used to explain the..”

Response: We have revised the sentence accordingly. Thank you

Figure 4, B: Unreadable

Response: Thank you for the comment. We have adjusted the figure accordingly.

Lines 233-235: Sentences are confusing as written.

Response: The sentence has been revised for clarity.

Figure 6: The y-axes are count of what? Explain. Also, explain what 2.5, 75, 375 and 1250 are in the table legend.

Response: Thank you for the comment. Counts per million (CPM) are obtained by normalizing the read counts by the total counts per sample.

2.5, 75, 375, and 1250 are the ammonium concentrations (treatments) used in the samples' below-ground part(root). We have clarified it in Table 1. Thank you

Line 256 and earlier: Explain what “upregulation” means.

Response: Thank you for the comment. Generally, upregulation is a phenomenon where genes are expressed at higher levels due to environmental stimuli, for example, in our case due to ammonium stress.

Conclusion: These conclusions do not conclude much. Provide an explanation for some of the results.

Response: Thank you for the comment. We have revised the section for clarity.
